# Left Ventricular Diastolic Indices and Their Impact on Outcomes in Patients with Recently Diagnosed Atrial Fibrillation

**DOI:** 10.3390/jcm11195732

**Published:** 2022-09-28

**Authors:** Nobuhiro Ikemura, Koki Nakanishi, John A. Spertus, Carolyn S. P. Lam, Takehiro Kimura, Yoshinori Katsumata, Taishi Fujisawa, Ikuko Ueda, Takahiro Ohki, Keiichi Fukuda, Seiji Takatsuki, Shun Kohsaka

**Affiliations:** 1Department of Cardiology, Keio University School of Medicine, Tokyo 160-8582, Japan; 2Kansas City Healthcare Institute for Innovations in Quality and Saint Luke’s Mid America Heart Institute, University of Missouri, Kansas City, MO 64111, USA; 3Department of Cardiovascular Medicine, The University of Tokyo, Tokyo 113-8654, Japan; 4Department of Cardiology, National Heart Center Singapore, Singapore 169609, Singapore; 5Department of Cardiology, Tokyo Dental College Ichikawa General Hospital, Ichikawa 272-8513, Japan

**Keywords:** atrial fibrillation, heart failure, diastolic dysfunction, quality of life

## Abstract

Background: Early identification of atrial fibrillation (AF) patients at risk for heart failure (HF) remains critical for improving their outcomes. We aimed to investigate whether indices of left ventricular diastolic dysfunction (LVDD) can stratify AF patients without clinical history of HF. Methods: We extracted 1775 patients’ data from a prospective cohort that consecutively recruited recently recognized AF patients with ejection fraction ≥50%. We categorized patients as LVDD grade 0 (none) to 3 (severe) based on mitral deceleration time and E/e’ per the American Society of Echocardiography recommendation. The primary outcome was a composite of all-cause death, stroke, and HF hospitalization during the 2-year follow-up. We also investigated the Atrial Fibrillation Effects on QualiTy-of-Life (AFEQT) scores. Results: Overall, 857 (48.3%) had mild or higher LVDD. Incidence of primary outcomes increased in parallel with LVDD grading (1.8%, 2.8%, 6.5%, and 8.1% for grades 0–3, respectively, *p* < 0.001), and the presence of grade 3 LVDD was an independent predictor of the primary outcome (adjusted HR 2.28 (vs. grade 0), 95%CI 1.13–4.60). Furthermore, patients with LVDD had lower AFEQT scores at the enrollment and 1-year follow-up. Conclusions: LVDD indices were associated with adverse clinical outcomes and patients’ perceived health status in a recently diagnosed AF cohort without HF.

## 1. Introduction

The incidence rates of atrial fibrillation (AF) and heart failure (HF) are increasing worldwide [1,2]. As the two conditions share common risk factors and have similar pathophysiology, approximately 20–30% of patients with either HF or AF tend to develop the other condition spontaneously, and a combined condition confers a greater mortality risk than either condition alone [3]. Therefore, identifying individuals with AF at high risk for progression to HF is of substantial clinical importance, as the AF-related treatments should be tailored to individual risk profiles [4,5,6].

Recent studies have indicated that AF could be an early manifestation of underlying left atrial failure, which leads to or is derived from left ventricular (LV) diastolic dysfunction (LVDD) [7,8]. Beyond understanding LV ejection fraction (EF), LVDD provides independent and incremental prognostic information on patients with established HF [9]. However, whether LVDD indices can predict risk in AF patients without a clinical history of HF is less well understood, in part because comprehensive LVDD assessments in AF are challenging (i.e., left atrial [LA] enragements frequently occur in AF patients regardless of filling pressures). Furthermore, the clinical implications of LVDD indices on health-related quality of life (HR-QoL) among AF patients remain unclear.

In the present analysis, we examined whether LVDD grade, based on two rhythm-independent indices, mitral deceleration time (DT) and E/e’ (by mitral inflow Doppler and tissue Doppler echocardiography), can help stratify patients at risk of adverse clinical outcomes, including HR-QoL, among newly diagnosed or referred AF patients without a clinical history of HF.

## 2. Materials and Methods

### 2.1. Data Sources

We used data from the Keio interhospital Cardiovascular Studies-atrial fibrillation (KiCS-AF) registry for this study. The rationale and design of the registry have been described previously [1,2]. Briefly, it is a prospective, multicenter, registry-based cohort study designed to collect data on the clinical variables and outcomes of consecutive AF patients between September 2012 to May 2018 from 11 hospitals within the Tokyo metropolitan area of Japan. The registry includes patients with prevalent or incident AF who are newly diagnosed or referred to the participating hospitals within the previous six months. Data on approximately 150 clinical variables of each patient’s background, symptoms, prior and current drug use, including oral anticoagulants, electrocardiography and echocardiography results, and blood sampling test results, were collected [2]. The Atrial Fibrillation Effect on Quality of Life (AFEQT) questionnaires were administered to all patients at the baseline visit and during 1-year follow-up visits or by mail if possible. Yearly follow-up examinations were performed for all patients by chart reviews, mail, and phone interviews. Dedicated study coordinators updated the status of major cardiovascular events and laboratory test results, performed procedures, recorded subsequent changes in the medications, and transcribed the AFEQT questionnaires [2]. Data quality assurance was achieved through systematic validation that highlighted outliers and data completeness, and the clinical research coordinators at each institution answered all queries regarding data entry. To ensure consecutive case enrollment, the senior study coordinator (I.U.) and investigator (S.K.) performed on-site auditing to ensure proper registration of each eligible patient [2]. The registry was conducted per the tenets of the Declaration of Helsinki. Institutional review board/ethics committee approval was obtained from all the study sites. All patients provided written informed consent [2].

### 2.2. Assessment of LVDD

We extracted data from patients who underwent echocardiography within three months before registration and had adequate records for LVDD assessment. The collected echocardiography variables comprised those obtained from the evaluation of LV function, including the quantification of LVEF via Simpson’s method in the 4-chamber and 2-chamber views; the LA diameter measured in the parasternal LV long-axis view; diastolic function measurements with mitral inflow Doppler and tissue Doppler echocardiography; and the presence or absence of valvular heart disease. To acquire echocardiographic parameters, at least 5 consecutive heartbeats were recorded and averaged for each parameter. During the echocardiographic examination, 52.5% of the patients (*n* = 932) had normal sinus rhythm (e.g., those with paroxysmal atrial fibrillation), 44.7% of the patients (*n* = 794) had AF and remains had pacemaker rhythm.

To assess LVDD comprehensively, we excluded patients with an LVEF of less than 50% and clinical history of HF. Then, we divided the remaining patients into two according to whether their average e’ ≥ 8 cm/s or not; we defined patients with e’ ≥ 8 cm/s as having no diastolic dysfunction (grade 0). Within patients with e’ < 8 cm/s, we calculated the individual LVDD score according to both the average E/e’ ratio and DT in all patients with e’ < 8 cm/s with prespecified cut-off values based on the American Society of Echocardiography (ASE) recommendations published in 2009 [3].

-A score of 1 was assigned if E/e’ ratio ≤8 or DT > 200 ms-A score of 2 was assigned if E/e’ ratio 9–12 or DT 160–200 ms-A score of 3 was assigned if E/e’ ratio ≥13 or DT < 160 ms

Consequently, the individual LVDD scores ranged from 2–6, and we graded the LVDD severity as follows: mild (grade 1) for those with individual LVDD scores of 2 or 3; moderate (grade 2) for those with an individual LVDD score of 4; and severe (grade 3) for those with individual LVDD scores of 5 or 6. (Appendix A) For example, patients with an e’ < 8 cm/s with an average E/e’ ratio ≥ 13 and a DT > 200 ms were assigned an individual LVDD score of 4 (grade 2 LVDD).

The present study did not include LA measurement within this grading system since LA enlargement is frequently observed within AF patients independent of the LV filling pressure [4]. Furthermore, trans-mitral E/A waves were not used in the grading of LVDD owing to the missing trans-mitral A wave in AF. All echocardiography reports were confirmed by board-certified cardiologists (the Japan Society of Ultrasonics in Medicine) in each institution.

### 2.3. Study Outcomes

For this analysis, the primary outcome measure was major adverse cardiovascular or neurological events (MACNEs), defined as a composite of all-cause death, stroke/non-central nervous system (CNS) systemic embolism, or new-onset HF hospitalization. While previous studies have shown that the combination of AF and HF further advance thrombogenicity in the left atrium [5], we included stroke/non-CNS systemic embolism as a composite outcome. In addition, the individual MACNE components were also assessed. All events were adjudicated by the endpoint adjudication committee, including three cardiologists, by reviewing health records and querying the clinical research coordinators responsible for each site.

### 2.4. Assessment of Patients’ Health Status

In the KiCS-AF, patients were requested to answer the internationally validated AFEQT questionnaire (http://www.afeqt.org, accessed on 3 September 2022) at registration and the 1-year follow-up visit or by mail. The AFEQT is a 20-item questionnaire that quantifies four domains of AF-related QoL, including symptoms, daily activities, treatment concerns, and treatment satisfaction, using 7-point Likert response scales [6]. An overall summary score can be calculated from the first three domains and ranges from 0–100, where 0 represents the most severe symptoms, physical limitations, and treatment concerns, and 100 represents the best AF-specific health status. A previous study comparing the EHRA symptom classification in AF and AFEQT showed that the mean AFEQT-OS score in patients classified as EHRA class 1 (e.g., no symptom) was 78.4 (SD 19.0) [7]. Thus, we regarded patients with AFEQT-OS scores ≥80 as those with preserved HR-QoL and patients with AFEQT-OS scores <80 as those with impaired HR-QoL. A culturally and linguistically translated version of the AFEQT for Japan was used.

### 2.5. Statistical Analysis

The baseline characteristics for the analytic cohort are presented as mean with SD for continuous variables and numbers with percentages for categorical variables. We compared the baseline characteristics across the LVDD grades using linear trend tests for continuous variables and Mantel-Haenszel trend tests for categorical variables. The time to MACNE development over 2 years was summarized using Kaplan–Meier estimates. This was then compared across the LVDD grades using a log-rank test. The Cox proportional hazards regression models were used to assess the association of LVDD grade and clinical variables with MACNEs during the 2-year after registration; the results are presented as hazard ratios (HRs) with their 95% confidence intervals (CIs). The model stratified for the patients within sites and was adjusted for the following clinically relevant factors: LVDD grade (grade 0 as reference), sex, age (as a contentious variable, hypertension, diabetes mellitus, prior stroke, or transient ischemic attack, coronary artery disease or peripheral artery disease, paroxysmal AF, use of oral anticoagulants at registration, body mass index (as a contentious variable), baseline estimated glomerular filtration rate (eGFR, as a continuous variable) and LA diameter (as a continuous variable). The rates of missing data for patient-level factors were all < 2%, except for eGFR (3.9%/*n* = 76). To account for missing data, we used a single median imputation. In addition, we performed sensitivity analyses following the exclusion of patients with moderate or severe mitral valve disease, left ventricular hyper trophy, and a cardiac pacemakers as these conditions affect diastology assessment and may be associated with their outcomes.

As for health status assessment, the AFEQT scores at baseline and 1-year follow-up were compared between patients with and without LVDD using the Student’s *t*-test, and results were illustrated using a radar chart [8]. In addition, changes in the AFEQT scores at 1 year from the baseline were compared between two groups using covariance analysis, adjusted for the baseline AFEQT-OS scores (e.g., a positive change represents improved HR-QoL and a negative change implies worsening). To investigate whether the incidence of LVDD was associated with impaired HR-QoL (AFEQT-OS scores < 80) at enrollment, we constructed a logistic regression model with generalized estimating equations to account for clustering of patients within sites and adjusted for aforementioned clinically relevant variables. Further, we performed a sub-group analysis for LA diameter (e.g., LA diameter < 40 mm or ≥40 mm), as LA dilatation is reportedly associated with worse HR-QoL among AF patients [9]. In order to ensure that we examined a representative cohort of patients, we examined differences in the baseline characteristics between those with and without available AFEQT data. IBM SPSS version 24.0 (IBM Corp., Armonk, NY, USA) was used for all the analyses. All reported *p*-values were two-sided, and a *p*-value < 0.05 was considered statistically significant.

## 3. Results

### 3.1. Baseline Characteristics

A total of 3166 consecutive AF outpatients were included in the registry, of whom 2247 (67.8%) underwent echocardiography within three months before registration and had adequate records for LVDD assessment (Figure 1).

Among these patients, 364 patients (16.2%) with a prior clinical diagnosis of HF were excluded. Furthermore, 108 (5.7%) with an LVEF less than 50% were excluded, as those with a reduced EF are known to have severe LVDD [4]. The remaining 1775 patients’ data was investigated in the present study. Overall, 70.2% (*n* = 1233) were men. The mean patient age was 66.3 (SD 11.2) years, and the mean CHA_2_DS_2_-VASc score was 2.08 (SD 1.4). The date of diagnosis of AF was documented in 90.3% of all patients (*n* = 1585), and of these, 62.5% (*n*= 1096) had less than one year between the diagnosis of AF and the study enrollment. Among these patients, 1711 patients (97.4%) had 2-year follow-up data for major cardiovascular events, and the mean follow-up period was 673 (standard deviation [SD] 160.9) days.

From the echocardiographic measurements, 918 (52.3%), 303 (17.2%), 299 (17.0%), and 255 (14.5%) patients had an LVDD severity grade of 0, 1, 2, and 3, respectively. (Figure 1). Table 1 presents a comparison of the demographic and clinical characteristics of the patients across the LVDD grades. Patients with higher-grade LVDD were more likely to be older, female, and have more comorbidities. There were no significant differences in the EF and LA diameter across these groups. Patients with higher-grade LVDD were more likely to be treated with diuretics, oral anticoagulants and less likely to have received antiarrhythmic drugs at the baseline (Table 1).

### 3.2. Composite Clinical Outcomes

During the follow-up, 63 patients experienced a MACNE. Figure 2 shows the Kaplan–Meier curves for MACNEs across the LVDD grades.

Log-rank test shows the significant difference in MACNE across all grades of left ventricular diastolic dysfunction (LVDD) at baseline (*p* < 0.001). Major adverse cardiovascular or neurological events (MACNE) was defined as a composite of all-cause death, stroke/non-CNS systemic embolism, and heart failure hospitalization.

The incidence of MACNEs increased in parallel to the degree of LVDD at the baseline (1.8%, 2.8%, 6.5%, and 8.1% for LVDD grades 0, 1, 2, and 3, respectively, *p* < 0.001, Table 2). After adjusting for known confounders, severe LVDD (grade 3) remained an independent predictor for MACNEs (adjusted HR 2.28 [with grade 0 as a reference group], 95%CI 1.13–4.60, *p* = 0.021, Table 2 and Appendix A). After excluding patients with moderate or severe mitral valve disease (5.2%/ *n* = 92), left ventricular hypertrophy (5.1%, *n* = 90), and a cardiac pacemakers (1.2%, *n* = 22) LVDD grade 2 remained an independent predictor of MACNEs (adjusted HR 2.38 [with grade 0 as a reference group], 95%CI 1.19–4.74, *p* = 0.01, Appendix A). In contrast, LVDD grade 3 was not an independent predictor.

### 3.3. Health Status Outcomes

As for health status outcomes, patients answered the AFEQT questionnaire at 1-year follow-up, on average, 420 (SD 73.0) days after enrollment. The rates of missing AFEQT data were 1.0% (*n* = 18/1775) at baseline and 13.8% (*n* = 245/1775) at 1 year. The characteristics of the patients without AFEQT data (*n* = 255, 14.4%) were largely comparable to those in the analytic cohort, although patients with missing data were less likely to have dyslipidemia, chronic kidney disease, and paroxysmal AF (Appendix A).

In the analytic cohort, patients with LVDD (at any grade) had worse AFEQT-OS scores at baseline than those without (75.2 ± 18.3 vs. 78.2 ± 16.7; *p* < 0.001, Table 3, Figure 3). The proportion of patients with impaired HR-QoL (AFEQT-OS score < 80) was higher among patients with LVDD than those without (51.7% vs. 42.3%; *p* < 0.001). After adjusting for clinically relevant variables, the incidence of LVDD was an independent predictor for patients with impaired HR-QoL at baseline (adjusted Odds ratio [OR] 1.38, 95%CI 1.12–1.71; *p* = 0.002, Appendix A).

At 1-year follow-up, patients had improved AFEQT scores regardless of LVDD (Figure 3), albeit patients with LVDD (at any grade) continued to have worse AFEQT-OS scores than those without (85.2 ± 14.4 vs. 87.0 ± 13.3, *p* = 0.002, Table 3). These differences were mainly driven by a lesser improvement in sub-scale for daily activities among patients with LVDD (mean change in sub-scale for daily activities within 1-year 6.8, 95%CI 5.7–7.9 vs. 9.0, 95%CI 7.9–10.1; *p* = 0.006, Table 3).

As for the sub-group analysis of LA dilatation, among patients without LA dilatation, those with LVDD at the baseline were less likely to have improved AFEQT-OS scores during the 1-year follow-up; this trend was predominantly driven by the poor improvement in the sub-scale for daily activities (Appendix A). However, among patients with LA dilatation, the degree of improvement in the AFEQT score was similar in magnitude between those with and without LVDD (Appendix A).

## 4. Discussion

In the present study, from a multicenter registry of newly recognized AF without a clinical history of HF, we found that almost half of all patients with a preserved EF had LVDD at baseline with a high incidence of all-cause death and hospitalization for HF. Furthermore, patients with LVDD had a worse HR-QoL at the baseline and 1 year after registration than those without LVDD. Notably, these differences were predominantly driven by impairment in daily activities and poor improvement on its during follow-up.

AF and HF are closely intertwined and associated with impaired heart performance, severe symptoms, and worsening QoL [10]. The assessment of their association is being pursued actively [11], whereas most investigations are from the HF perspectives. The present result indicates that LVDD is also a significant risk factor for developing HF, even in patients with AF, in line with recent comprehensive concepts for common underlying atrial and ventricular myopathy [10]. The incident HF following the diagnosis of AF was 4% in our cohort, and this finding was comparable to those observed in the Outcomes Registry for Better Informed Treatment of Atrial Fibrillation (ORBIT-AF) [12]. A recent report from the longitudinal cohort study indicates that cardiovascular events within AF patients are triggered early in the disease course [13], and this is also consistent with the relatively higher incidence of HF in early recognized AF observed in our study. The large-scale clinical trials have indicated that the selected sodium-glucose cotransporter 2 inhibitors prevent the development of HF among patients with HF with preserved EF [14]. Further observational studies have reported that weight loss and intensive risk factor control in patients with AF have favorable effects on cardiac structure and function that may reduce incident HF [15]. Collectively, these findings represent an essential breakthrough in the treatment of diastolic HF patients. Further researches are needed to identify preventive and therapeutic strategies to effectively reduce the risk of developing HF in patients with AF.

To the best of our knowledge, this is the first study describing the association between HR-QoL and LVDD in AF patients and demonstrating that LVDD is related to a worse HR-QoL, especially for daily activities. LVDD is reportedly associated with a reduced exercise capacity due to reduced myocardial relaxation rates, decreased LV suction force, and elevated ventricular filling pressure [16]; these adverse effects may impact patients’ reported HR-QoL. In addition, the trends in patients’ reported HR-QoL across the LVDD grades varied with LA dilatation status. Therefore, assessing LVDD with LA diameter may help predict HR-QoL in AF patients.

The results from our study indicate that LVDD indices, based on two rhythm-independent indices (E/e’ and DT), could be used as a predictor of future adverse clinical outcomes in patients at the early stage of AF. A user-friendly bedside pre-screening tool would be beneficial for the managing clinicians. Notably, our results indicate that the LVDD remains a significant predictor regardless of the LA diameter, suggesting that assessment of LA diameter alone is insufficient in the risk prediction of AF patients. LA size might reflect adverse electrophysiological or structural changes in the atrial myocardial tissue in AF rather than LVDD severity [4].

Our study has several potential limitations. First, this is a non-randomized observational study with inherent limitations; however, this design is most suitable for describing the current treatment patterns and outcomes, as it is impossible to randomize patients to different degrees of LVDD. Nevertheless, unmeasured confounding cannot be excluded, potentially due to depression, frailty, or economic status. Furthermore, even with statistical adjustment, it is not possible to completely adjust for confounding factors (e.g., patients with LVDD had much more cardiovascular risk factors). Second, we did not perform echocardiography as part of the study protocol (although this was for clinical reasons). Moreover, standardized echocardiographic protocols, specified echocardiography machines, and follow-up echocardiography were not mandatory in the KiCS-AF registry. However, board-certified cardiologists confirmed all the echocardiography findings and measurements for patients with inadequate image quality were not included in the registry. Third, the assessment of LVDD grade in the present study was based on the cut-off value extracted from the 2009 recommendation for evaluating LV diastolic function by the ASE [3]. Although it has recently been updated [4], since the registry was formulated in September 2012, the variables were fixed at that time. In addition, a recent study demonstrated that applying the new 2016 ASE/ European Association of Cardiovascular Imaging recommendations for diastolic function assessment revealed a much lower prevalence of LVDD; the latest classification might be capable of detecting only the most advanced cases [17]. Fourth, the present study did not include LA measurement within the LVDD grading since LA enlargement is frequently observed within AF patients independent of the LV filling pressure. Nevertheless, further investigations to explore the generalizability of our findings for another AF population are warranted since the registry primarily included patients with early stages of AF with modest LA enlargement. Fifth, we defined LA dilatation by the LA diameter measured in the parasternal LV long axis or 4-chamber views and not by the LA volume. Finally, not all AF patients in Japan participated in the KiCS-AF registry [2]. Sampling bias and the level of generalizability of our findings across Japan are potential concerns, although we had complete enrollment from the 11 centers in the registry. Regardless of these concerns, the present study, which included one of the most representative Japanese databases of AF patients, presents a complete assessment of Japan’s current practice patterns and HR-QoL outcomes.

## 5. Conclusions

In contemporary practice, recently diagnosed AF patients with preserved EF, baseline LVDD assessment by two rhythm-independent indices, the average E/e’ and mitral deceleration time, was favorable to predict clinical outcomes, including perceived health status. Measuring LVDD indices in the clinical care of patients with AF can aid in providing updated prognostic information.

## Figures and Tables

**Figure 1 jcm-11-05732-f001:**
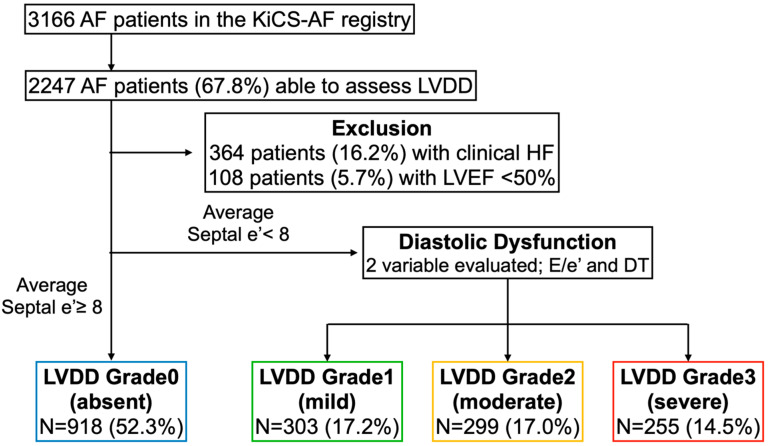
Study flow chart. Abbreviations; AF, atrial fibrillation; KiCS-AF, Keio Interhospital Cardiovascular Studies-Atrial Fibrillation; LVDD, left ventricular diastolic dysfunction; LVEF, left ventricular ejection fraction, DT, deceleration time.

**Figure 2 jcm-11-05732-f002:**
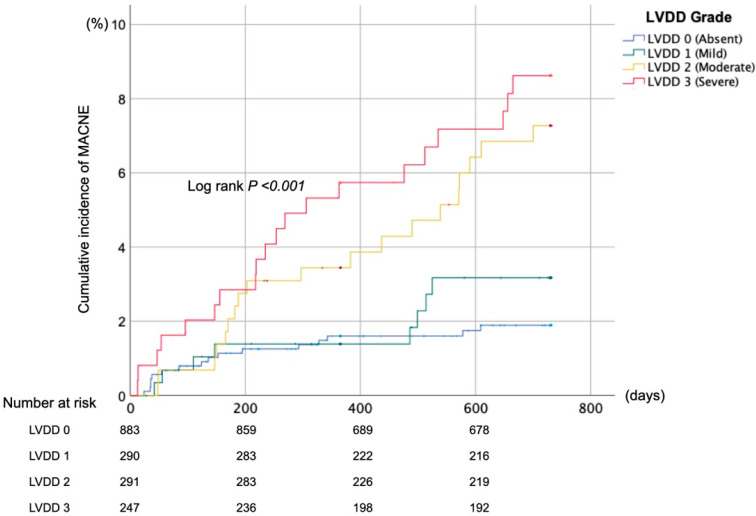
Kaplan–Meier curves for cumulative incidence of major adverse cardiovascular or neurological events during 2-year follow-up.

**Figure 3 jcm-11-05732-f003:**
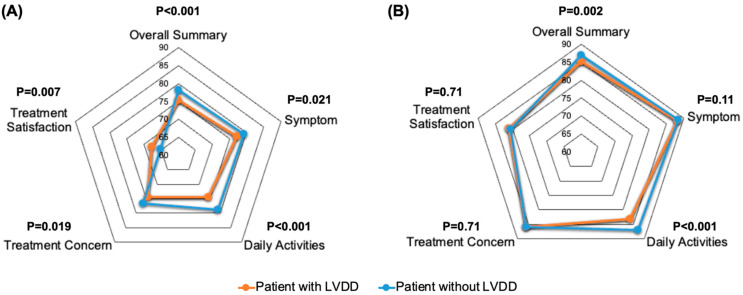
The Atrial Fibrillation Effects on QualiTy-of-Life outcomes at baseline and 1-year follow-up. Comparison of the Atrial Fibrillation Effects on QualiTy-of-Life overall summary score and individual domains in patients with and without left ventricular diastolic dysfunction (LVDD) at (**A**) enrollment and (**B**) 1 year after registration. Each dot represents mean AFEQT scores, and *p* values indicate differences between the two groups.

**Table 1 jcm-11-05732-t001:** Baseline characteristics of the analytic cohort.

Characteristics, *n* (%)	Patients with Normal Diastolic Function (Grade 0) *n* = 918	Patients with Mild Diastolic Dysfunction (Grade 1) *n* = 303	Patients with Moderate Diastolic Dysfunction (Grade 2) *n* = 299	Patient with Severe Diastolic Dysfunction (Grade 3) *n* = 255	*p* Value
Age, mean, years (SD)	62.8 (11.5)	68.8 (9.2)	70.3 (9.8)	71.6 (9.7)	<0.001
Men	683 (74.4)	207 (68.3)	188 (62.9)	155 (60.8)	<0.001
BMI, median, kg/m^2^ (SD)	23.8 (3.5)	23.5 (3.4)	23.5 (3.6)	23.5 (3.3)	0.159
Heart rate, mean, bpm (SD)	79.4 (17.6)	73.7 (15.0)	76.9 (16.0)	79.8 (18.9)	0.052
Blood pressure, mean, mmHg (SD)					
Systolic	130.2 (18.8)	134.2 (19.8)	133.4 (18.4)	133.3 (18.5)	0.002
Diastolic	79 (13.2)	76.8 (13.5)	76.8 (13.5)	77.4 (12.5)	0.039
Medical history					
Smoking	155 (16.9)	46 (15.2)	44 (14.7)	28 (11.0)	0.13
Hypertension	419 (45.6)	189 (62.4)	196 (65.6)	168 (65.9)	<0.001
Diabetes mellitus	103 (11.2)	42 (13.9)	50 (16.7)	52 (20.4)	<0.001
Dyslipidemia	257 (28.0)	122 (40.3)	124 (41.5)	97 (38.0)	<0.001
Stroke or TIA	56 (6.1)	28 (9.2)	29 (9.7)	23 (9.0)	0.084
CKD (eGFR < 60 mL/min)	297 (32.4)	121 (39.9)	121 (40.5)	123 (48.2)	<0.001
Peripheral artery disease	18 (2.0)	12 (4.0)	12 (4.0)	10 (3.9)	0.099
Coronary artery disease	27 (2.9)	15 (5.0)	21 (7.0)	24 (9.4)	<0.001
Pacemaker implantation	5 (0.5)	4 (1.3)	8 (2.7)	5 (2.0)	0.021
Echocardiographic Parameters					
Ejection Fraction, % (SD)	60.2 (3.7)	60.1 (3.5)	60.3 (3.8)	60.2 (3.8)	0.28
LV Hypertrophy	33 (3.6)	11 (3.6)	27 (9.0)	19 (7.5)	<0.001
LA diameter, mm (SD)	40 (7)	37 (6)	40 (7)	41 (7)	0.20
Average e’, cm/s (SD)	10.3 (2.0)	6.4 (1.1)	5.9 (1.3)	6.3 (1.2)	<0.001
E/e’ ratio (SD)	8 (2.4)	9.2 (2.0)	13.5 (5.1)	15 (5.4)	<0.001
Deceleration time, msec (SD)	181.3 (48.5)	243.6 (48.3)	210.6 (49.7)	151.7 (26.1)	0.61
Moderate or severe MS	0	0	1(0.1)	0	0.34
Moderate or severe MR	37 (4.0)	6 (2.0)	21 (7.0)	28 (11.0)	<0.001
Type of visit					
Referral from emergency department	46 (5.0)	33 (10.9)	29 (9.7)	26 (10.2)	<0.001
Diagnosed at health screening	336 (36.6)	53 (17.5)	63 (21.1)	61 (23.9)	<0.001
Type of AF at registration					
First detected	39 (4.2)	20 (6.6)	21 (7.0)	11 (4.3)	<0.001
Paroxysmal	434 (47.3)	249 (82.5)	194 (64.9)	109 (42.9)	
Persistent	291 (31.7)	21 (7.0)	58 (19.4)	87 (34.3)	
Permanent	136 (14.8)	10 (3.3)	25 (8.4)	45 (17.7)	
Current drug therapy					
β-blockers	430 (46.8)	131 (43.2)	161 (53.8)	142 (55.7)	0.005
ACE inhibitors/ARBs	239 (26.0)	121 (39.9)	124 (41.5)	97 (38.0)	<0.001
Calcium-channel blockers	330 (35.9)	127 (41.9)	137 (45.8)	134 (52.5)	<0.001
Digoxin	27 (2.9)	5 (1.7)	10 (3.3)	10 (3.9)	0.42
Diuretics	60 (6.5)	18 (5.9)	31 (10.4)	37 (14.5)	<0.001
Currently using antiarrhythmic drugs	216 (23.5)	101 (33.3)	80 (26.8)	41 (16.1)	<0.001
Oral anticoagulants					
None	190 (20.7)	56 (18.5)	47 (15.7)	16 (6.3)	<0.001
Warfarin	86 (9.4)	28 (9.2)	29 (9.7)	37 (14.5)	0.099
Direct oral anticoagulants	643 (70.0)	219 (72.3)	223 (74.6)	202 (79.2)	0.027
Prior interventional therapy for AF					
Catheter ablation of AF	50 (5.4)	37 (12.2)	31 (10.4)	25 (9.8)	<0.001
Surgical maze	1 (0.1)	0 (0.0)	2 (0.7)	1 (0.4)	0.24
BNP, mean, pg/mL, (SD)	105.3 (107.5)	77 (123.9)	139.7 (162.2)	171.9 (137.9)	<0.001
CHA_2_DS_2_-VASc score, (SD)	1.6 (1.4)	2.3 (1.4)	2.6 (1.5)	2.7 (1.3)	<0.001

Abbreviations: SD, standard deviation; BMI, body mass index; TIA, transient ischemic attack; CKD, chronic kidney disease; eGFR, estimated glomerular filtration rate; PCI, percutaneous coronary intervention; LVEF, left ventricular ejection fraction; LA, left atrium; MS, mitral stenosis; MR, mitral regurgitation; ACE, angiotensin-converting enzyme; ARB, angiotensin receptor blocker; BNP, brain natriuretic peptide.

**Table 2 jcm-11-05732-t002:** Clinical outcomes across patients with diastolic dysfunction.

Outcomes		Patients with Normal Diastolic Function (Grade 0) *n* = 918	Patients with Mild Diastolic Dysfunction (Grade 1) *n* = 303	Patients with Moderate Diastolic Dysfunction (Grade 2) *n* = 299	Patient with Severe Diastolic Dysfunction (Grade 3) *n* = 255	*p* Value
MACNE	Incidence, *n* (%)	16 (1.8)	8 (2.8)	19 (6.5)	20 (8.1)	<0.001
	Adjusted HRs (95%CI)	Reference	1.10 (0.44–2.76)	1.82 (0.89–3.71)	2.28 (1.13–4.60)	-
All-cause death	Incidence, *n* (%)	3 (0.3)	3 (1.0)	10 (3.4)	5 (2.0)	<0.001
	Adjusted HRs (95%CI)	Reference	2.46 (0.44–13.5)	4.66 (1.21–17.9)	2.27 (0.50–10.1)	-
Heart failure hospitalization	Incidence, *n* (%)	9 (1.0)	2 (0.7)	5 (1.7)	11 (4.5)	0.001
	Adjusted HRs (95%CI)	Reference	0.39 (0.07–2.03)	0.71 (0.21–2.34)	1.78 (0.68–4.69)	-
Stroke	Incidence, *n* (%)	5 (0.6)	3 (1.0)	4 (1.4)	1 (0.4)	0.66
	Adjusted HRs (95%CI)	Reference	1.35 (0.27–6.59)	1.32 (0.32–5.47)	0.48 (0.05–4.42)	-

MACNE, major adverse cardiovascular or neurological events defined as a composite of all-cause death, stroke/non-CNS systemic embolism, and heart failure hospitalization. Each model stratified for the patients within sites, and was adjusted for clinically relevant factors as follows; LVDD grades (grade 0 as reference), sex, age (as contentious variables), hypertension, prior stroke or transient ischemic attack, coronary artery disease or peripheral artery disease, paroxysmal AF, use of oral anticoagulants at baseline, body mass index (as contentious variables), baseline estimated glomerular filtration rate (eGFR; as a continuous variable) and LA diameter (per 1 cm increase).

**Table 3 jcm-11-05732-t003:** The Atrial Fibrillation Effects on QualiTy-of-Life outcomes across patients with diastolic dysfunction.

	Patients with Normal Diastolic Function (Grade 0) *n* = 918 (51.7%)	Patients with Diastolic Dysfunction (Any Grade) *n* = 857 (48.3%)	*p* Value
Baseline, mean (SD)			
Overall summary	78.2 (16.7)	75.2 (18.3)	<0.001
Symptom	79.2 (18.8)	77.1 (19.9)	0.021
Daily activities	78.8 (20.7)	74.5 (22.8)	<0.001
Treatment concerns	76.6 (17.5)	74.5 (19.2)	0.019
Treatment satisfaction	65.1 (20.8)	67.7 (20.1)	0.007
1-year after registration, mean (SD)			
Overall summary	87.0 (13.3)	85.2 (14.4)	0.002
Symptom	88.6 (14.8)	88.3 (15.5)	0.11
Daily activities	87.0 (16.8)	83.2 (18.4)	<0.001
Treatment concerns	85.8 (13.4)	85.9 (14.7)	0.71
Treatment satisfaction	80.7 (19.2)	81.2 (18.0)	0.71
Change within 1-year, mean (95% confidence interval) *			
Overall summary	9.6 (8.7–10.4)	8.5 (7.7–9.4)	0.10
Symptom	10.2 (9.5–11.2)	9.6 (8.6–10.6)	0.38
Daily activities	9.0 (7.9–10.1)	6.8 (5.7–7.9)	0.006
Treatment concerns	10.1 (9.2–11.0)	10.6 (9.6–11.5)	0.50
Treatment satisfaction	14.5 (13.0– 15.9)	14.6 (13.1–16.1)	0.90

* Changes in AFEQT score within 1 year were defined as AFEQT score at 1-year minus AFEQT score at baseline and were compared between each group by using analysis of covariance adjusted for baseline AFEQT scores. A positive change represents improved HR-QoL, and a negative change implies worsening HR-QoL.

## Data Availability

The data and materials used to conduct this research are available to researchers, on request, for scientific projects aimed at identifying a novel clinical finding that may further improve patient outcome. Attempts to co-validate country-specific observations, risk stratification schemes, and outcomes are also welcome. The procedure does need to follow the Act on the Protection of Personal Information Law (as of May 2017) and the Ethical Guidelines for Medical and Health Research Involving Human Subjects (as of March 2021) in Japan.

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
