# Peer review of "Left Ventricular Diastolic Indices and Their Impact on Outcomes in Patients with Recently Diagnosed Atrial Fibrillation"

_jcm, 2022, doi:10.3390/jcm11195732_

Round 1
Reviewer 1 Report
1) Please append information how the group size was calculated.
2) Were the echo results verified or cross-checked during the study? Mean LA diameter result is very small and range narrow all in all groups (40-41 mm), even in the group with 11% of moderate to severe MR and ongoing AFib – please comment on that as it not often seen in clinical practice.
3) Please provide the number of patients with inadequate image quality to perform EF and LVDD measurements.
4) Patients with no diastolic disfunction had higher BNP and larger LA diameter than subjects with mild diastolic disfunction – please comment on that.
Reviewer 2 Report
In the present work (Left Ventricular Diastolic Indices and Teir Impact on Outcomes in Patients with Recently Diagnosed Atrial Fibrillation), authors expose an interesting study where they review the influence of Left Ventricular Diastolic Disfunction (LVDD) in patients with preserved Left Ventricular Ejection Fraction and newly diagnosed Atrial Fibrillation. It is a valuable paper, well written, with updated bibliography and with some quite interesting results.
Although authors present quite an interesting paper, there are still some issues that if addressed, may improve their work.
1#. Left Atrial enlargement is excluded for grading patients with LVDD. Authors correctly affirm that it is highly present in patients with LVDD, but this is not a reason to be excluded. Although In table 1 is shown that it is equally distributed among all groups, authors should include a statement in limitations, since it is not known how it would affect in patient classification.
#2. Patients with LVDD are older and have more Cardiovascular risk factors (hypertension, diabetes mellitus, dyslipidemia, CKD and Coronary A. Disease). Although this factors are included in the multivariate analysis, authors should still include this in the limitations section, since this is an observational study and it is not known how it would affect the results.
Thanks for letting me review this article.
Good luck in your further investigations.
Round 2
Reviewer 1 Report
All issues were clarified.